# Hop Tropicalization: Chemical Compositions of Varieties Grown under Organic and Conventional Systems in Subtropical Conditions

Gabriel Cássia Fortuna [1], Caio Scardini Neves [1], Olivia Pak Campos [1], Jordany Aparecida Oliveira Gomes [1], Júlio César Rodrigues Lopes Silva [1], Amauri Alves Souza [1], Cristiano Soleo de Funari [1], Márcia Ortiz Mayo Marques [2] and Filipe Pereira Giardini Bonfim [1,*]

[1] Department of Plant Production, School of Agronomic Sciences, Botucatu Campus, Júlio de Mesquita Filho São Paulo State University, Botucatu 18610-034, SP, Brazil; gabriel.cassia.fortuna@gmail.com (G.C.F.); caioscardini@gmail.com (C.S.N.); olivia.pakc@gmail.com (O.P.C.); jordanyoliveira327@gmail.com (J.A.O.G.); julio.lopes@unesp.br (J.C.R.L.S.); amaurialvejunior@gmail.com (A.A.S.); cristiano.funari@unesp.br (C.S.d.F.)

[2] Center for Research on Plant Genetic Resources, Agronomic Institute, Avenida Barão de Itapura, 1481, Botafogo, Campinas 13020-902, SP, Brazil; marcia.marques@sp.gov.br

* Correspondence: filipe.giardini@unesp.br

**Abstract:** The interest in the production of hops in Brazil, motivated by the third position in the world ranking of beer producers and the growth of the craft brewery business, justifies the intensification of studies into its adaptation to local growing conditions. Due to the high internal demand, the aim of this study was to evaluate the phytochemical profiles of hop varieties grown in subtropical conditions under different cropping systems. Studies that promote the expansion of cultivation areas in distinct climate conditions and ensure quality are very important. A randomized block design was adopted with a 2 × 5 subdivided plot. The main factor was the cropping system (organic and conventional), and the secondary factor was the hop variety (Columbus, Chinook, Nugget, Cascade and Hallertau Mittelfrüeh), with four blocks and four plants per plot. The quality parameters monitored in this work were the contents of alpha and beta acids, and xanthohumol in the inflorescences of hops, as well as the relative composition of their essential oils. The variations in the chemical profiles of essential oils showed differences between some varieties, and the different compositions and levels resulting from the two cropping systems show that management and cultural practices can influence the aromatic characteristics of hops; in total, 23 compounds were found. The terpene fraction represented 79.67% of the oil in Hallertau and 93.63% in Cascade, with myrcene being the main compound. The levels of bitter acids and xanthohumol did not differ statistically as a function of the treatments. This study contributes the first records of the chemical profiles of hops grown in subtropical conditions in Brazil, in general, the Nugget variety had the highest qualitative potential

**Keywords:** alpha and beta acids; Brazilian hops; essential oil; management; phytochemical profile; xanthohumol

## 1. Introduction

Hop (*Humulus lupulus* L.) is a perennial, herbaceous and dioecious vine that is considered a horticultural plant due to its agricultural suitability and multiple uses. The hop plant produces inflorescences annually; lupulin glands develop in these structures, also called cones, and biosynthesize specialized metabolites such as terpenoids, alpha and beta acids, and phenolic compounds, among other substances whose properties characterize hop quality [1,2].

In brewing, hop essential oils contribute to beer aroma and flavor, which can confer a range of notes (woody, citrus, spicy, floral, fruity, sulfurous, tangy, herbal, resinous, and earthy), according to the chemical profile of a cone [3]. Alpha acid is related to bitterness,

with values ranging from 2–8% for aromatic varieties to 12–18% for bitter varieties [4]. Beta acids are less important to the brewing process; however, they present high levels of antimicrobial activity due to the presence of three isoprenyl groups that act as antioxidant and preservative agents [5].

Xanthohumol is a relevant hop polyphenol that has been widely studied as a prenylflavonoid that can act as an inhibitor in the initiation, promotion and progression of carcinogenesis [6].

Brazil is the third-largest beer producer in the world; however, it stands out as the largest importer of hop in South America. In 2019, Brazil imported 3.24 thousand tons of hop at a cost of approximately US $57 million [7] because hop is predominantly cultivated in temperate climates. There is potential to cultivate this crop in other regions due to the many varieties adapted to different climatic conditions [8]. Different performances, compositions and levels of metabolites are found within and among varieties since plant expression depends on interactions with external edaphoclimatic and biotic factors [9].

Climatic and geographical characteristics, plant genetics, age and health, environment, plant interactions, cropping conditions, cultural practices, management, and postharvest practices are sources of variability in hop cone chemical composition [10–14]. In this sense, the cropping systems can influence productivity and quality, the basic differences between organic and conventional systems are the fertilization sources and plant protection protocols, which according to Grzyb et al. [15] affect the composition of plants. Solarska and Sosnowska [16] report in their studies that some hop varieties perform better under organic cropping systems than conventional systems.

Clinical studies with small animals support that consumption of organically produced food is better for human health than conventionally produced [16]. There is a growing concern with nutrition linked to health, in addition to the alarming necessity maintaining environmental sustainability, and also the economic interest. The conventional agricultural practices use levels of inputs that can result in a disruption of the natural production of specialized metabolites in the plants, so, this management affect the nutrients levels in plants [15].

Therefore, in view of the high internal demand and expansion of the national brewing market with interest in hops with peculiar phytochemical profiles [17], studies that promote the expansion of new cultivation zones and that ensure quality are very important. Thus, the aim of this study was to evaluate the chemical profiles of essential oils and alpha and beta acids and xanthohumol contents of five hop varieties cultivated in subtropical conditions (Botucatu-SP, Brazil) in organic and conventional cropping systems.

## 2. Materials and Methods

### 2.1. Experimental Area

The experiment was conducted at the Department of Horticulture of the School of Agronomic Sciences of UNESP in the municipality of Botucatu-SP, Brazil (latitude, 22°50′ S; longitude, 48°26′ W; elevation, 791 m). According to Köppen [18], the climate is classified as subtropical with hot summers (Cfa), and the soil in the study area is clayey dystroferric Red Latosol. The hops were planted in November 2018, and the research data were collected in the second year of production (November 2019 to March 2020). In this period, the minimum average temperature was 17.94 °C, the maximum average temperature was 28.45 °C, and the cumulative rainfall was 1257.61 mm. The annual average minimum and maximum temperatures were 15.83 °C and 25.91 °C, respectively, and the annual average precipitation was 100.23 mm.

### 2.2. Treatments and Experimental Design

A randomized block design was adopted with a 2 × 5 split plot; the main factor was the cropping systems (organic and conventional) and the secondary factor was the hop varieties (Columbus, Chinook, Nugget, Cascade and Hallertau Mittelfrüeh), with four blocks and four plants per plot. The organic and conventional management systems

were differentiated mainly by fertilization and phytosanitary control, following Brazilian legislation and the recommendations established for hop in international literature [19]. The elements used in each cropping system are described in the next section.

### 2.2.1. Description of Varieties

As described above, the varieties analyzed in this research were Columbus, Chinook, Nugget, Cascade and Hallertau Mittelfrüeh. They were used in this research because they have characteristics that distinguish them from each other, such as their profile and contents of essential oils, or that generate distinct aromatic characteristics and percentages of alpha and beta acids. These factors determine the potential styles of beers that can be produced with each variety. Finally, the varieties have characteristics that determine whether they are more or less bitter and have a more intense or lighter aromatic profile. Therefore, each variety is characterized by its unique characteristics, a fact that justifies the development of new varieties and their commercial planting.

Columbus is a variety that has dual aptitude, and its alpha acid level and percentage of essential oils characterizes it for use as a bittering or aroma hop. Its aroma is pungent with citrus notes. The alpha and beta acid levels of this variety vary between 14 and 18% and 4.5 and 6%, respectively, and the total essential oil contents vary between 1.5 and 4.5 mL/100 g [20].

Chinook also has dual functions, as it has a high load of alpha acids (12–14%) and is widely used to provide bitterness to beers; however, due to the composition of its oils (1.5–2.7 mL/100 g) and its aroma, which is characterized by pine and spices, it is also used to provide aroma in certain styles of beers [20,21].

Nugget is a variety that has an intense herbal aroma, light flavor and marked bitterness and is used both to provide aroma and bitterness to beers, so it also has dual functions; it has approximately 9.5–14% alpha acids and 1.5–3 mL/100 g total essential oils [20,22].

Cascade is one of the most popular and widely cultivated varieties in the world due to its excellent development and vigor. It has dual functions but is most commonly used to provide aroma to beers, and its aromatic profile has floral notes with citrus and grapefruit elements. The alpha acid content is lower, from 4.5–8.9%, and therefore, it is not commonly used to provide bitterness to beers, and its total oils are between 0.8 and 1.5 mL/100 g [20,21,23].

Hallertau is one of the varieties that is considered noble; it has been cultivated for more than 100 years in Germany. It is used only for providing aroma to beers and has a slightly floral and spicy aromatic characteristic. The contents of the total essential oils is between 0.6 and 2 mL/100 g, and the alpha acid content is 3.5% [20,24].

Thus, each variety has unique characteristics that allow the use of its cones in different styles of beers. For example, Columbus is commonly used in more distinctive styles, such as imperial stout, stout and American ales; Chinook in pale ale, India pale ale, porter, stout, lager, American lager and others; Cascade mainly in American pale ale, ale and lager; and finally, Hallertau in German pilsner, pale ale, wheat and American lager [20].

Several styles of beer are produced with each variety, and the number of uses for each variety can increase further since there is no fixed rule about the use of a certain hop variety for a particular style of beer, leaving it open to the creativity of the brewmaster.

### 2.2.2. Conventional Cultivation

Fertilizer was applied according to the needs determined by soil analyses (Table A1). In the first year, topdressing containing calcium nitrate (375 kg ha$^{-1}$), urea (94 kg ha$^{-1}$), potassium chloride (186 kg ha$^{-1}$) and micronutrients with MIB$^{®}$ (20 kg ha$^{-1}$) was added. Phytosanitary control included applications of abamectin (Abamex$^{®}$) for streaked mites (*Tetranychus urticae*), fipronil (Regent$^{®}$) for leaf-cutting ants, and tebuconazole (Folicur$^{®}$) for powdery mildew (*Podosphaera macularis*), which was identified in the first year. In the second year, the same fertilizers were used, and conventional poultry litter (3.12 t ha$^{-1}$) was added. Borate fertilization was done with boric acid (4 kg ha$^{-1}$), and leaf fertilization

was done with zinc sulfate (5 kg ha$^{-1}$). The phytosanitary control of mites and ants was identical to that described above, and *Bacillus thuringiensis* (Dipel$^{®}$) was applied following the appearance of caterpillars.

### 2.2.3. Organic Farming

Fertilizer was applied according to the needs determined by soil analyses (Table A1). In the first year of cultivation, cattle manure, castor bean cakes and potassium sulfate were used. For phytosanitary control, sulfur–calcium spray solutions were applied for streaked mites (*T. urticae*), organic formicides (Bioisca$^{®}$) for ants, and raw milk and Bordeaux mixture for powdery mildew (*P. macularis*). In the second year, fertilization was done with bokashi (1.5 t ha$^{-1}$), castor bean cakes (1.4 t ha$^{-1}$), and organic poultry litter (2 t ha$^{-1}$). Potassium sulfate (94 kg ha$^{-1}$), potassium silicate (312 kg ha$^{-1}$), thermophosphate (203 kg ha$^{-1}$), boric acid (4 kg ha$^{-1}$) and bone meal (1 t ha$^{-1}$) were also used. Spraying was performed with SuperMagro biofertilizer, and biological activation of the soil was performed with effective microorganisms (EM). *Metarhizium anisopliae + Beuaveria bassiana* (B Ex-change$^{®}$) were applied for preventative pest control.

### 2.3. Evaluations

#### 2.3.1. Chemical Composition of the Essential Oil

The hop samples analyzed were collected when the cones reached the mature stage (February–March 2020). The plants were harvested in full and taken to the laboratory to remove the cones; these were dried in a forced air ventilation oven at 40 °C for a variable time from 24 to 48 h until they reached approximately 10% moisture.

The extraction of essential oils was performed by hydrodistillation in a Clevenger apparatus from 50 g cones for 1 h and 30 min.

The determination of the essential oil chemical profiles was performed at the Instituto Agronônico de Campinas (IAC) in a gas chromatograph coupled to a mass spectrometer (CG-EM, QP 5000–Shimadzu) equipped with an OV-5 MS capillary column and helium as the capillary gas.

The system was operated in full scan mode with electron impact (70 eV), and ranged from 40 to 450 m/z. The injector was kept at 220 °C, with a carrier gas flow rate of 1:20 and temperature programming of 60 °C–240 °C (3 °C min$^{-1}$). The interface temperature was maintained at 240° C. Oil samples were diluted in ethyl acetate, and 1 µL of solution was injected.

For quantitative analysis, a gas chromatograph with a flame ionization detector (CG-DIC, Shimadzu, CG-2010/AOC-20i) was used. The system was equipped with an OV-5 capillary column, helium as the carrier gas, injector at 280 °C, detector at 300 °C, 1:20 split and the same temperature program as the GC-MS system.

The identification of chemical constituents was performed by comparing the mass spectra of the substances with the National Institute of Standards and Technology library (Nist 62.lib) and the substance retention indices [25]; these indices were obtained from the injection of a mixture of n-alkanes (C9–C24, Sigma, St. Louis, MO, USA) under the same chromatographic conditions as the samples, applying the equation by Van Den Dool and Kratz [26].

#### 2.3.2. Quantification of Alpha Acids, Beta Acids and Xanthohumol

Quantifications were based on methods reported by Prencipe [27]. Exactly 50 mg of ground cones were extracted by dynamic maceration at 1000 rpm for 30 min (Heidolph MR Hei-Tec, Germany) with 2.0 mL of MeOH-HCOOH (99:1 *v:v*). After filtration (22 µm PTFE syringe filter), 1.5 µL was injected into an ultrahigh-performance liquid chromatograph coupled to a UV/Vis spectrophotometer (Shimadzu Nexera UC, Kyoto, Japan). Separations were achieved in a C18 column of 150 × 2.1 mm and 1.7 µm (Kinetex, Phenomenex, Torrance, CA, USA). The mobile phase was composed of $H_2O$ and acetonitrile (ACN), both with 0.25% HCOOH, in the following gradient: 35–75% ACN (0–12 min), 75–100% ACN

(12–25 min), and 100% ACN (25–28 min). The analysis temperature and flow rate were 30 °C and 0.39 mL min$^{-1}$, respectively. Online UV spectra were recorded from 200 to 400 nm. Data were processed using LabSolutions LCMS software version 5.96 (Shimadzu Corporation, Japan).

### 2.4. Statistical Analysis

The chemical composition data of the essential oils were assessed with multivariate principal component analysis (PCA) using Minitab® Statistical Software [28]. Alpha and beta acid and xanthohumol data were subjected to analysis of variance, and means were compared using the Scott–Knott test at 5% significance using Sisvar software [29].

### 3. Results

In this study, the chemical compositions of hop varieties grown in a subtropical environment in Brazil under organic and conventional systems were elucidated for the first time.

### 3.1. Chemical Composition of the Essential Oils

The chemical composition of the essential oils is described in Table 1; the volatile fraction showed variations mainly in terms of relative percentages, and in total, 23 substances were identified. The main classes were monoterpenes, oxygenated monoterpenes, sesquiterpene hydrocarbons, oxygenated sesquiterpenes, esters, aldehydes and ketones; the main volatile components were myrcene, linalool, caryophyllene, farnesene and ledene.

The terpene fraction was relatively large, accounting for 79.67% of Con Hal essential oil and 96.63% of Org Cas essential oil, with myrcene being the major compound in all varieties in both cropping systems, except for Con Hal, whose most abundant terpene compound was beta-farnesene (38.43%). Org Hal had a 62% higher myrcene content (39.26%) than the same variety grown under the conventional system (14.83%); furthermore, Con Hal showed a myrcene content that was 71% lower than Con Cas (51.63%).

Linalool is the most abundant oxygenated terpene alcohol; the highest content of this compound was found in Con Nug (0.91% ± 0.26) and the lowest content was found in Org Chi (0.52% ± 0.19).

Beta-caryophyllene had a higher percentage in the organic cropping system than in the conventional system, with averages of 6.04% and 3.79%, respectively; among the varieties, Columbus had the highest content of this compound, being 77%, 38%, 29% and 20% higher than the contents in Hallertau Mittelfrüeh, Cascade, Chinook, and Nugget, respectively.

Beta-farnesene was the sesquiterpene present at the highest percentage in the five varieties; in Hallertau Mittelfrüeh, this content of this compound was higher than in other varieties, with an average content of 38.43% in the conventional system. This variety presented 12.27% more beta-farnesene than Con Nug, which presented the second highest percentage (26.16% ± 8.78) and 12.59% more than Org Hal (38.43% ± 2.33).

The sesquiterpenes ledene and beta-selinene stood out among the chemical compositions of the varieties, with percentages ranging from 6.70% (Org Hal) to 11.72% (Org Col) and 6.77% (Org Hal) to 10.76% (Org Col), respectively.

Through PCA (Figure 1), it was possible to verify the similarities between the varieties in the cropping systems. Figure 1 shows the PCA, with 76.80% of the total variance explained by the first two principal components. The main difference was observed for Hallertau Mittelfrüeh, the chemical characteristics of which were distinguished from the other varieties in both cropping systems; in particular, the presence of substances from the ketone and aldehyde groups differentiated this variety. The substances methyl heptanone, 2-nonanal, n-nonanal, methyl octanoate, 2-undecanone, undecanal, undec-9E-em-1-al, 2-methyl-lavandula butanoate, germacrene B and eudesmol were observed only in the Hallertau Mittelfrüeh variety.

**Table 1.** Essential oil chemical constituents of *Humulus lupulus* L. varieties grown in conventional and organic cropping systems under subtropical conditions.

| | Substance (%) | LRI Cal | LRI Lit | Org Cas | Con Cas | Org Nug | Con Nug | Org Chi | Con Chi | Org Hal | Con Hal | Org Col | Con Col |
|---|---|---|---|---|---|---|---|---|---|---|---|---|---|
| 1 | β-Pinene | 976 | 974 | 0.90 ± 0.06 | 0.68 ± 0.32 | 0.48 ± 0.30 | 0.52 ± 0.37 | 0.57 ± 0.39 | 0.82 ± 0.02 | 0.68 ± 0.06 | 0.23 ± 0.00 | 0.50 ± 0.18 | 0.50 ± 0.19 |
| 2 | Myrcene | 991 | 988 | 55.92 ± 7.21 | 51.63 ± 19.08 | 41.40 ± 13.49 | 43.20 ± 16.43 | 43.97 ± 19.06 | 57.65 ± 2.68 | 39.26 ± 3.15 | 14.83 ± 5.65 | 38.27 ± 11.82 | 44.84 ± 3.83 |
| 3 | Methylheptanone | 1021 | 1021 | -- | -- | -- | -- | -- | -- | 0.38 ± 0.11 | 0.27 ± 0.07 | -- | -- |
| 4 | 2-Nonanal | 1089 | 1087 | -- | -- | -- | -- | -- | -- | 0.43 ± 0.14 | 0.28 ± 0.10 | -- | -- |
| 5 | Linalool | 1098 | 1095 | 0.65 ± 0.08 | 0.79 ± 0.27 | 0.62 ± 0.13 | 0.91 ± 0.26 | 0.52 ± 0.19 | 0.68 ± 0.04 | 0.62 ± 0.12 | 0.66 ± 0.00 | 0.59 ± 0.25 | 0.72 ± 0.11 |
| 6 | n-Nonanal | 1102 | 1100 | -- | -- | -- | -- | -- | -- | 0.63 ± 0.34 | 0.54 ± 0.13 | -- | -- |
| 7 | Methyl octanoate | 1123 | 1123 | -- | -- | -- | -- | -- | -- | 0.24 ± 0.04 | 0.23 ± 0.02 | -- | -- |
| 8 | 2-Decanone | 1191 | 1192 | 0.24 ± 0.01 | 0.32 ± 0.18 | -- | -- | -- | -- | 1.09 ± 0.92 | 0.49 ± 0.05 | -- | -- |
| 9 | Methyl nonanoate | 1222 | 1223 | 0.60 ± 0.26 | 1.03 ± 0.60 | 0.40 ± 0.19 | 1.39 ± 0.60 | 0.45 ± 0.15 | 0.69 ± 0.32 | 0.56 ± 0.22 | 1.16 ± 0.18 | 0.23 ± 0.00 | 0.23 ± 0.00 |
| 10 | 2-Undecanone | 1191 | 1293 | -- | -- | -- | -- | -- | -- | 0.64 ± 0.55 | 2.42 ± 0.31 | -- | -- |
| 11 | Undecanal | 1307 | 1305 | -- | -- | -- | -- | -- | -- | 0.56 ± 0.22 | 1.16 ± 0.18 | -- | -- |
| 12 | Undec-9E-em-1-al | 1312 | 1311 | -- | -- | -- | -- | -- | -- | 0.60 ± 0.25 | 1.47 ± 0.30 | -- | -- |
| 13 | (E)-Caryophyllene | 1417 | 1417 | 4.99 ± 1.85 | 3.82 ± 1.24 | 7.49 ± 1.85 | 3.89 ± 0.88 | 7.20 ± 4.07 | 2.81 ± 0.15 | 2.13 ± 0.34 | 2.60 ± 0.18 | 8.37 ± 1.52 | 5.85 ± 1.05 |
| 14 | α-(E)-Bergamotene | 1434 | 1432 | 0.78 ± 0.10 | 0.93 ± 0.40 | 1.00 ± 0.23 | 1.11 ± 0.37 | 0.97 ± 0.30 | 0.80 ± 0.05 | 1.43 ± 0.27 | 1.89 ± 0.15 | 1.04 ± 0.22 | 0.98 ± 0.11 |
| 15 | (E)-β-Farnesene | 1456 | 1452 | 17.24 ± 1.16 | 22.85 ± 9.31 | 23.98 ± 5.52 | 26.16 ± 8.78 | 22.69 ± 6.75 | 19.16 ± 1.10 | 25.84 ± 2.12 | 38.43 ± 2.33 | 25.29 ± 5.66 | 24.01 ± 2.65 |
| 16 | β-Charmigrene | 1474 | 1476 | 1.23 ± 0.25 | 1.23 ± 0.49 | 1.67 ± 0.40 | 1.63 ± 0.45 | 1.63 ± 0.59 | 1.09 ± 0.09 | 1.23 ± 0.20 | 1.87 ± 0.19 | 1.76 ± 0.35 | 1.56 ± 0.11 |
| 17 | β-Selinene | 1485 | 1489 | 7.37 ± 1.45 | 7.40 ± 2.95 | 10.05 ± 2.43 | 9.73 ± 2.77 | 9.72 ± 3.42 | 6.63 ± 0.62 | 6.77 ± 1.15 | 10.56 ± 1.22 | 10.66 ± 2.07 | 9.43 ± 0.75 |
| 18 | Ledene | 1494 | 1496 | 8.08 ± 1.57 | 8.12 ± 3.24 | 11.09 ± 2.57 | 10.65 ± 2.99 | 10.72 ± 3.86 | 7.41 ± 0.67 | 6.70 ± 1.03 | 9.26 ± 0.57 | 11.72 ± 2.31 | 10.19 ± 0.65 |
| 19 | 2-Methyl-lavandula butanoate | 1513 | 1511 | -- | -- | -- | -- | -- | -- | 0.74 ± 0.17 | 0.77 ± 0.13 | -- | -- |
| 20 | α-Cadinene | 1532 | 1537 | 0.30 ± 0.08 | -- | 0.46 ± 0.16 | | | 0.33 ± 0.20 | 0.37 ± 0.11 | 0.81 ± 0.14 | 1.22 ± 0.13 | 0.27 ± 0.08 | -- |
| 21 | mi | 1554 | 1559 | -- | | | | | | 0.81 ± 0.13 | 1.20 ± 0.09 | | |
| 22 | α-Eudesmol | 1650 | 1652 | -- | -- | -- | -- | -- | -- | 0.88 ± 0.20 | 1.49 ± 0.20 | -- | -- |
| 23 | 6-Z-Pentadecen-2-o | 1667 | 1667 | 0.63 ± 0.32 | 0.68 ± 0.23 | 1.04 ± 0.27 | 0.82 ± 0.25 | 1.00 ± 0.43 | 0.55 ± 0.04 | 0.90 ± 0.26 | 1.47 ± 0.23 | 0.94 ± 0.17 | 0.87 ± 0.11 |
| | Monoterpene hydrocarbons | | | 56.82 | 52.31 | 41.88 | 43.72 | 44.54 | 57.68 | 39.94 | 15.06 | 38.77 | 45.34 |
| | Oxygenated monoterpenes | | | 0.65 | 0.79 | 0.62 | 0.91 | 0.52 | 0.68 | 0.62 | 0.66 | 0.59 | 0.72 |
| | Sesquiterpene hydrocarbons | | | 39.81 | 44.35 | 55.28 | 53.17 | 53.27 | 38 | 44.10 | 64.61 | 59.94 | 52.02 |
| | Oxygenated sesquiterpenes | | | -- | -- | -- | -- | -- | -- | 0.88 | 1.49 | -- | -- |
| | Esters | | | 0.60 | 1.03 | 0.40 | 1.39 | 0.45 | 0.69 | 1.18 | 1.66 | 0.23 | 0.23 |
| | Ketones | | | -- | -- | -- | -- | -- | -- | 2.22 | 3.45 | -- | -- |
| | Aldehydes | | | 0.24 | 0.32 | -- | -- | -- | -- | 1.73 | 2.91 | -- | -- |

LRI Cal = Calculated linear retention index; LRI Lit = linear retention index stated in the literature. (Org Cas—Organic Cascade, Con Cas—conventional Cascade, Org Nug—organic Nugget, Con Nug—conventional Nugget, Org Chi—organic Chinook, Con Chi—conventional Chinook, Org Hal—organic Hallertau Mittelfrüeh, Con Hal—conventional Hallertau Mittelfrüeh, Org Col—organic Columbus, Con Col—conventional Columbus).

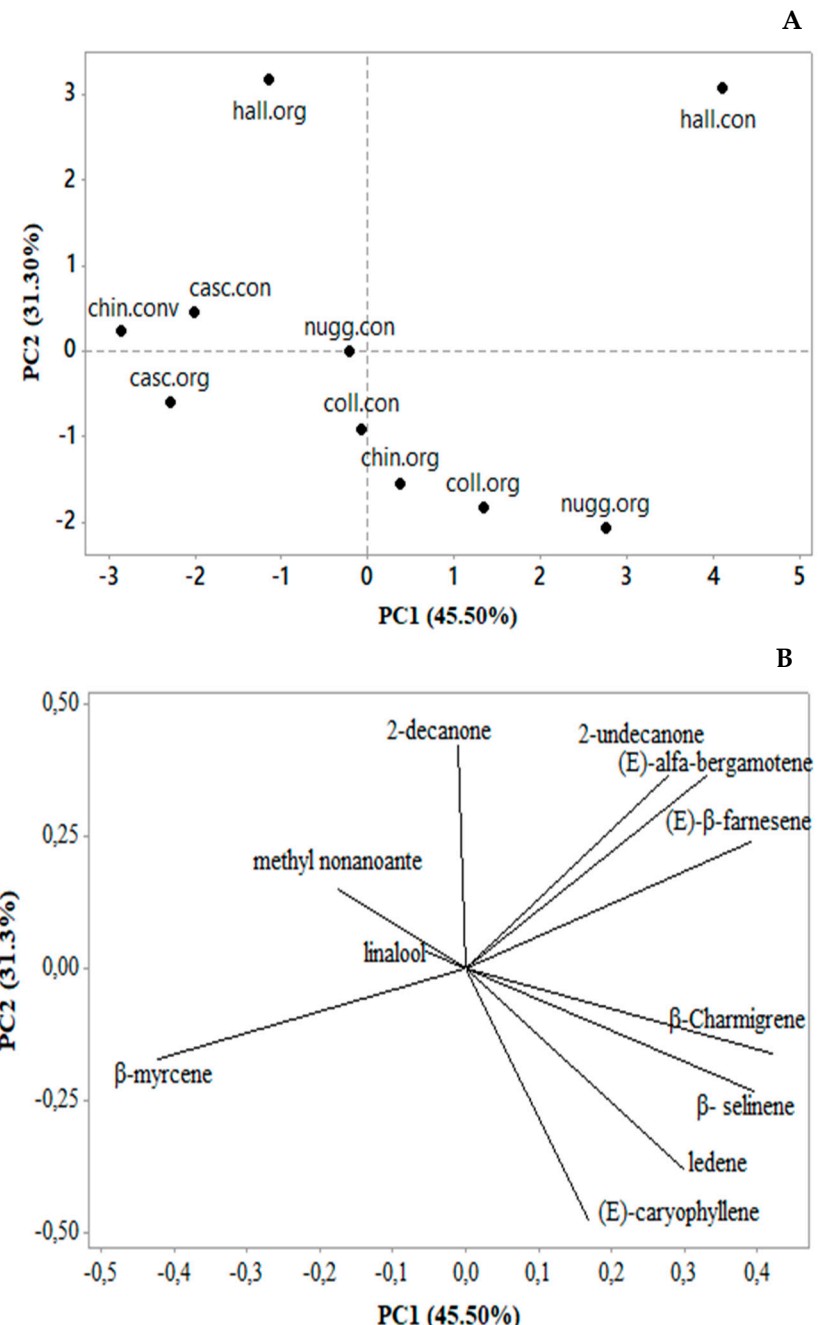

**Figure 1.** Plots of the principal component analysis of *Humulus lupulus* L. varieties cultivated in organic and conventional cultivation systems and their chemical constituents. Score plot (**A**) and loadings plot (**B**). See Table 1 for trait labels.

The Cascade, Chinook, Nugget and Columbus varieties showed greater similarity; however, small variations within varieties were observed based on the cropping system. The monoterpene myrcene showed the highest correlation with Cascade grown in both systems and Chinook grown in the conventional system.

### 3.2. Alpha acids and Beta Acids Contents

The fixed fraction of inflorescences of *H. lupulus* is mainly composed of xanthohumol and bitter acids (alpha and beta acids). The major alpha acids in these materials are cohumulone, humulone and adhumulone, while the major beta acids are colupulone, lupulone and adlupulone [30]. Therefore, these six bitter acids and xanthohumol were

quantified here and summarized in Table 2. The total concentration of bitter acids ranged from $1.08 \pm 0.21\%$ (Org Col) to $6.16 \pm 6.93\%$ (Org Nug), and the total xanthohumol content ranged from $0.38 \pm 0.03$ mg g$^{-1}$ (Org Cas) to $1.57 \pm 1.40$ mg g$^{-1}$ (Org Nug).

**Table 2.** Alpha and beta acids and xanthohumol contents of *Humulus lupulus* L. varieties grown in conventional and organic cropping systems under subtropical conditions.

| Substance | Org Cas | Con Cas | Org Nug | Con Nug | Org Chi | Con Chi | Org Hal | Con Hal | Org Col | Con Col |
|---|---|---|---|---|---|---|---|---|---|---|
| n-Humulone (mg g$^{-1}$) | 0.77 ± 1.42 | 0.98 ± 1.26 | 8.17 ± 9.53 | 2.98 ± 1.81 | 1.38 ± 1.51 | 0.87 ± 1.56 | 2.39 ± 0.60 | 3.19 ± 2.33 | 0.69 ± 1.49 | 2.22 ± 3.09 |
| Cohumulone (mg g$^{-1}$) | 2.86 ± 0.92 | 3.46 ± 1.72 | 28.78 ± 32.47 | 10.13 ± 7.79 | 3.45 ± 0.13 | 3.16 ± 0.78 | 8.38 ± 3.88 | 10.43 ± 5.02 | 2.34 ± 0.32 | 6.62 ± 7.64 |
| Adhumulone (mg g$^{-1}$) | 0.68 ± 0.21 | 0.91 ± 0.35 | 6.08 ± 2.78 | 2.04 ± 1.66 | 0.85 ± 0.03 | 0.76 ± 0.19 | 1.84 ± 0.90 | 2.38 ± 1.24 | 0.58 ± 0.10 | 2.06 ± 1.75 |
| Colupulone (mg g$^{-1}$) | 4.79 ± 0.94 | 6.78 ± 2.58 | 8.41 ± 6.31 | 6.27 ± 3.50 | 4.38 ± 0.39 | 4.41 ± 1.70 | 2.48 ± 0.95 | 3.18 ± 1.41 | 4.10 ± 0.92 | 6.31 ± 3.57 |
| n-Lupulone (mg g$^{-1}$) | 2.68 ± 1.09 | 3.64 ± 0.88 | 7.48 ± 5.50 | 3.70 ± 0.84 | 2.37 ± 1.00 | 2.41 ± 0.87 | 2.46 ± 0.47 | 3.03 ± 1.97 | 2.08 ± 1.31 | 3.31 ± 1.60 |
| Adlupulone (mg g$^{-1}$) | 1.26 ± 0.39 | 1.84 ± 0.47 | 2.69 ± 1.80 | 1.67 ± 0.68 | 1.19 ± 0.37 | 1.20 ± 0.39 | 0.89 ± 0.18 | 1.12 ± 0.73 | 1.01 ± 0.58 | 1.63 ± 0.87 |
| Alpha acids (%) | 0.43 ± 0.68 | 0.54 ± 0.59 | 4.31 ± 4.85 | 1.51 ± 0.70 | 0.53 ± 0.73 | 0.48 ± 0.78 | 1.26 ± 0.24 | 1.60 ± 1.15 | 0.36 ± 0.72 | 1.06 ± 1.39 |
| Beta acids (%) | 0.87 ± 0.26 | 1.23 ± 0.33 | 1.86 ± 1.23 | 1.16 ± 0.45 | 0.79 ± 0.27 | 0.80 ± 0.26 | 0.58 ± 0.11 | 0.73 ± 0.49 | 0.72 ± 0.38 | 1.12 ± 0.56 |
| Alpha + beta acids (%) | 1.31 ± 0.33 | 1.76 ± 0.69 | 6.16 ± 6.93 | 2.68 ± 1.25 | 1.32 ± 0.03 | 1.28 ± 0.17 | 1.84 ± 0.82 | 2.33 ± 1.11 | 1.08 ± 0.21 | 2.28 ± 1.84 |
| Xanthohumol (mg g$^{-1}$) | 0.38 ± 0.03 | 0.56 ± 0.20 | 1.57 ± 1.40 | 0.66 ± 0.08 | 0.79 ± 0.97 | 0.52 ± 0.23 | 0.72 ± 0.27 | 1.05 ± 0.31 | 0.44 ± 0.21 | 0.40 ± 0.05 |

(Org Cas—Organic Cascade, Con Cas—conventional Cascade, Org Nug—organic Nugget, Con Nug—conventional Nugget, Org Chi—organic Chinook, Con Chi—conventional Chinook, Org Hal—organic Hallertau Mittelfrüeh, Con Hal—conventional Hallertau Mittelfrüeh, Org Col—organic Columbus, Con Col—conventional Columbus).

## 4. Discussion

This work showed, for the first time, the characteristics of different hop varieties in subtropical conditions (Botucatu, SP-Brazil, latitude 22°50′ S) under conventional and organic cropping systems, focusing on specialized metabolites associated with the sweetening characteristics of hop cones for beer production.

### 4.1. Chemical Composition of the Essential Oils

Myrcene is a monoterpene and is considered the primary compound of hop essential oils [31]; because it is the major compound, myrcene levels may vary to a greater degree than those of other compounds. It is responsible for the aroma of green hops and is related to resinous, pine and pungent flavors, providing interesting flavors for the preparation of fruity beers and IPAs [32,33].

The values of myrcene in Cascade were within the standard range of 45 to 60% in both cropping systems (Org Cas 55.92, Con Cas 51.63), those in Chinook were in the range of 35 to 40% in both systems (Org Chi 43.97, Con Chi 57.65), those in Hallertau Mittelfrüeh ranged from 20 to 28% only in the organic system (Org Hal 39.26, Con Hal 14.83), those in Columbus reached the standard range of 25 to 40% in the organic system (38.27) and exceeded this range in the conventional system (44.84), and those in Nugget did not reach the range of 48 to 59% in either system (Org Nug 41.40, Con Nug 43.20) [34]. The low myrcene content observed in Con Hal was expected because this compound appears in smaller quantities in European hop varieties, such as Hallertau Mittelfrüeh [35].

The autooxidation of myrcene gives rise to several cyclic reaction products (e.g., alpha-pinene, beta-pinene, camphene, and r-cymene) and forms terpenoids such as linalool, nerol, geraniol, citral, alpha-terpineol and carvone [12]. Linalool and beta-pinene were observed in this study and may have been formed by the oxidation of myrcene; it is emphasized that a wide range of distinct aromas and flavors can be obtained with the compounds originating from myrcene.

Linalool is one of the most important indicators of beer aroma quality and hop freshness and is among the most interesting oxygenated aromatic compounds for the brewing market, with great sensory activity even though it is normally present in essential oil in

proportions smaller than 1% [32]. In the present study, Con Nug stood out among the varieties for having a higher linalool content.

Alpha-humulene, along with beta-caryophyllene, is the most abundant sesquiterpene in hops and it has positive impacts on beer aroma, being desired in a 3:1 ratio (alpha-humulene: beta-caryophyllene) to provide a more refined aromatic character with a strong emphasis on herbal, floral and spicy notes [32]. Alpha-humulene was not detected in this study, which may be related to the tropical cultivation conditions.

Increases in the percentage of beta-caryophyllene are commonly related to attack by pests and pathogens [36], justifying the higher percentage of this compound in the organic system, in which preventive control was limited, with management instead focused on maintaining insect populations at acceptable levels that did not result in crop damage.

Beta-farnesene, the sesquiterpene that was present at the highest percentage in all the evaluated varieties, especially Con Hal, can reach up to 30% of the total essential oils in noble hops, which are widely used in Bohemian pilsner style beers [32].

The ledene and beta-selinene contents contributed to the high contents of terpene hydrocarbons observed in this study, which exceeded the values reported by studies performed in temperate climates, which ranged from 50 to 80% [32,37].

Despite the influence of major compounds, the entire set of essential oils determines the formation of aromas of interest to the brewing industry. In hops, the presence of terpene hydrocarbons and substances such as esters, alcohols and ketones provide different aromatic ranges. In this context, the PCA showed similarities between the varieties in both cropping systems.

The differences observed may have occurred due to the environment and management practices (e.g., fertilization and phytosanitary control). These factors differed from those established in traditional cultivation sites and exerted a great influence on the hop essential oil chemical composition [38,39], thus allowing the expression of Brazilian "terror" in the chemical composition of the essential oils.

Organic cropping adds value to the product, in addition, brings environmental [40] and social benefits, and, as observed in this study and confirmed by others [15,41], can increase the levels of some chemical compounds of interest to the brewing and medicinal market, i.e., it is economically interesting.

### 4.2. Alpha Acid and Beta Acid Contents

Overall, all varieties cultivated in this work presented lower average levels of bitter acids than those described in the literature for *H. lupulus* cultivated in temperate zones (Table 2) (refer to "Description of varieties" subsection in the Materials and methods). It is also important to highlight that no statistical differences were observed among varieties or cropping systems in this work. This was expected due to the high relative standard deviations (RSD) observed among replicates for bitter acid contents (Table 2). As the HPLC-UV method used here was a validated method [27], being its reproducibility further confirmed in our laboratory from four commercial hop pellets (all with RSD $\leq$ 4.4%), it was concluded that the observed RSD evidence the high variability among specimens belonging to the same variety and cropping system. Both, the average low levels of bitter acids and the observed high RSD, can be at least partially explained by the fact that the evaluated plants were in their second year of cultivation. That is because the expected physiological maturation of plants should occur only in the third to fifth year of cultivation, when the biochemical machinery for the production of bitter acid tends to me more efficient and stable [42,43]. On the other hand, it is important to monitor the contents of such compounds throughout the development of the plants as carried out in this work, since it can give important feedbacks regarding the development of the plants and indicate tendencies.

Finally, the lower average levels of bitter acids found here compared to those found in temperate zones might be related to what was reported by Mozny et al. [44], who found that the increasing in temperatures observed in recent years in Czech Republic is inducing early flowering of the 'Saaz' hop variety, with reduced levels of alpha acids. Early flowering was

observed for the plants investigated here, probably due to both warmer days and shorter photoperiods found in Botucatu city when compared to traditional temperate cultivation sites [44].

## 5. Conclusions

The cultural practices and management adopted in this work altered the composition of hop volatile compounds. The same could not be concluded for bitter acids due to the high relative standard deviation found between the analyzes of the same field replicates. However, this evidenced a high variability among specimens belonging to the same variety and cultivation system regarding the production of bitter acids. As a consequence, such a high variability could mask any possible effect from the different managements and varieties of hops adopt in this work on the production of bitter acids. As it can be related with the fact that the evaluated plants were only in their second year of cultivation, new studies aiming to know their phytochemical evolution throughout their physiological maturation are being carried. Such studies are necessary to eventually establish a scientific basis that would allow the expansion of new areas of cultivation of hops in Botucatu city, which hosts several craft breweries that are highly dependent on hop imports from other countries.

**Author Contributions:** Conceptualization, G.C.F.; Data curation, G.C.F., J.A.O.G., C.S.d.F., M.O.M.M. and F.P.G.B.; Formal analysis, J.A.O.G., J.C.R.L.S., A.A.S., C.S.d.F., M.O.M.M. and F.P.G.B.; Funding acquisition, G.C.F., C.S.N., O.P.C., J.A.O.G. and F.P.G.B.; Investigation, G.C.F., C.S.N., O.P.C., J.A.O.G., J.C.R.L.S., A.A.S., M.O.M.M. and F.P.G.B.; Methodology, G.C.F., C.S.N. and O.P.C.; Project administration, G.C.F. and F.P.G.B.; Software, J.A.O.G.; Supervision, C.S.d.F., M.O.M.M. and F.P.G.B.; Validation, F.P.G.B.; Visualization, G.C.F.; Writing—original draft, G.C.F., J.A.O.G. and F.P.G.B.; Writing—review & editing, O.P.C. and F.P.G.B. All authors have read and agreed to the published version of the manuscript.

**Funding:** The authors are grateful to the Fundação de Amparo à Pesquisa do Estado de São Paulo (FAPESP) for financial support to implement and maintain the experimental area and for the purchase of equipment and inputs for analyses (2018 01786-1 and 2019/27066-8) and to the Coordenação de Aperfeiçoamento de Pessoal de Nível Superior (CAPES) and Conselho Nacional de Desenvolvimento Científico e Tecnológico (CNPq) for financial support through postgraduate scholarships.

**Conflicts of Interest:** The authors declare no conflict of interest.

## Appendix A

**Table A1.** Complete soil chemical analysis of organic and conventional cropping systems of *Humulus lupulus* L. between November 2018 and March 2020, Botucatu-SP.

| | | pH | OM | P | K | Ca | Mg | CEC | V% | S | B | Cu | Fe | Mn | Zn |
|---|---|---|---|---|---|---|---|---|---|---|---|---|---|---|---|
| | | CaCl$_2$ | G·dm$^{-3}$ | mg·dm$^{-3}$ | | mmol$_c$·dm$^{-3}$ | | | % | | | mg·dm$^{-3}$ | | | |
| **Date** | **Sys./Depth (cm)** | | | | | | | | | | | | | | |
| Nov. 2018 | Org 0–20 | 5.4 | 29 | 49 | 13.5 | 32 | 15 | 84 | 72 | 124 | 0.44 | 4.7 | 34 | 6.9 | 3.0 |
| | Org 20–40 | 5.0 | 22 | 35 | 7.5 | 25 | 12 | 84 | 53 | 77 | 0.63 | 4.9 | 34 | 5.3 | 1.6 |
| | Conv 0–20 | 5.6 | 19 | 29 | 9.6 | 26 | 9 | 65 | 68 | 139 | 0.38 | 5.4 | 27 | 4.7 | 0.6 |
| | Conv 20–40 | 5.3 | 18 | 12 | 2.6 | 14 | 5 | 51 | 42 | 52 | 0.31 | 5.5 | 39 | 4.6 | 0.4 |
| Apr. 2019 | Org 0–20 | 5.3 | 17 | 27 | 1.9 | 25 | 11 | 63 | 60 | 4 | 0.43 | 2.0 | 34 | 6.8 | 1.3 |
| | Org 20–40 | 5.0 | 17 | 23 | 2.9 | 14 | 11 | 67 | 42 | 25 | 0.55 | 1.2 | 34 | 5.7 | 1.1 |
| | Conv 0–20 | 5.4 | 22 | 34 | 1.4 | 29 | 10 | 70 | 58 | 27 | 0.48 | 1.9 | 35 | 4.5 | 0.5 |
| | Conv 20–40 | 4.5 | 16 | 10 | 1.2 | 12 | 7 | 63 | 33 | 43 | 0.44 | 0.5 | 36 | 2.1 | 0.4 |
| Aug. 2019 | Org 0–20 | 5.7 | 21 | 14 | 1.6 | 33 | 12 | 64 | 72 | 17 | 0.34 | 3.6 | 18 | 2.1 | 1.2 |
| | Org 20–40 | 4.4 | 15 | 3 | 0.5 | 11 | 5 | 64 | 26 | 67 | 0.33 | 4.5 | 18 | 0.8 | 0.1 |
| | Conv 0–20 | 5.0 | 19 | 17 | 0.3 | 19 | 7 | 58 | 45 | 34 | 0.30 | 4.1 | 21 | 2.4 | 0.2 |
| | Conv 20–40 | 4.3 | 15 | 4 | 0.4 | 11 | 4 | 70 | 21 | 48 | 0.35 | 4.8 | 17 | 1.7 | 0.2 |

**Table A1.** *Cont.*

| Date | Sys./Depth (cm) | pH CaCl₂ | OM G·dm⁻³ | P mg·dm⁻³ | K | Ca | Mg | CEC | V% % | S | B | Cu | Fe | Mn | Zn |
|---|---|---|---|---|---|---|---|---|---|---|---|---|---|---|---|
| | | | | | mmol_c·dm⁻³ | | | | | mg·dm⁻³ | | | | | |
| Nov. 2019 | Org 0–20 | 6.0 | 24 | 39 | 4.3 | 47 | 13 | 84 | 76 | 101 | 0.39 | 4.2 | 24 | 3.7 | 3.9 |
| | Org 20–40 | 5.6 | 22 | 40 | 2.8 | 40 | 17 | 83 | 71 | 80 | 0.63 | 3.8 | 24 | 3.4 | 3.4 |
| | Conv 0–20 | 5.8 | 25 | 44 | 4.0 | 80 | 10 | 119 | 79 | 422 | 0.63 | 3.6 | 19 | 3.4 | 3.3 |
| | Conv 20–40 | 5.6 | 25 | 49 | 5.5 | 66 | 8 | 107 | 74 | 305 | 0.58 | 4.0 | 26 | 3.9 | 2.8 |
| Mar. 2020 | Org 0–20 | 5.0 | 25 | 56 | 3.4 | 39 | 11 | 77 | 70 | 71 | 1.00 | 4.7 | 19 | 5.1 | 3.8 |
| | Org 20–40 | 4.8 | 19 | 30 | 2.2 | 23 | 10 | 70 | 50 | 52 | 0.90 | 5.4 | 17 | 3.0 | 1.7 |
| | Conv 0–20 | 4.9 | 19 | 16 | 2.6 | 20 | 5 | 64 | 42 | 42 | 1.09 | 6.1 | 19 | 3.2 | 0.9 |
| | Conv 20–40 | 4.4 | 15 | 6 | 1.4 | 13 | 4 | 63 | 29 | 148 | 0.86 | 5.3 | 16 | 3.1 | 0.4 |

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
