# Peer review of "Hop Tropicalization: Chemical Compositions of Varieties Grown under Organic and Conventional Systems in Subtropical Conditions"

_horticulturae, doi:10.3390/horticulturae9080855_

Round 1

Reviewer 1 Report

After reading the manuscript entitled: Tropicalization of Humulus lupulus L. (Cannabaceae): Chemical compositions of varieties grown under organic and conventional cropping system in subtropical conditions, Brazil, by Gabriel Cássia Fortuna, Caio Scardini Neves, Olivia Pak Campos, Jordany Aparecida Oliveira Gomes, Júlio César Rodrigues Lopes Silva, Amauri Alves Souza, Cristiano Soleo de Funari, Márcia Ortiz Mayo Marques, Filipe Pereira Giardini Bonfim I make some considerations.

Introduction: I would add something about conventional and organic types of cultivation and why to compare them as this is still not very clear;

Materials and Methods: Items 2.3.1 and 2.3.2. should have their paragraphs reduced considering that it makes it difficult to read and understand the paragraph.

Item 2.4....using Minitab statistical software (reference); ...significance using Sisvar (reference);

Results: In item 3.1 specify Figure 1 A and B, what each one represents. In the caption of figure 1 it would be better to specify what each graph shows. Suggestion: Principal component analysis of Humulus lupulus L. varieties cultivated in organic and conventional cultivation systems (A) and chemical constituents (B).

Discussion is the weakest point.  I think the discussion was lacking, rather than explaining each observed component. What are the differences between the types of management? Did the management used to allow the production of fruits with the chemical composition of the essential oil in the expected pattern? What this might mean for the country's market, industry, etc.)

item 4.1 ...Nugget reached the range of 48 to 59% in organic system (and what is the value in conventional systems, by way of comparison).

Conclusion: make a conclusion responding to the objective, succinctly.

Appendix A is not referenced throughout the text.

English needs proofreading

Author Response

The article has been thoroughly reviewed by a third party, and the English language and grammar have been improved.

Reviewer 2 Report

       Hops (Humulus lupulus L.) are a perennial plant of the Cannabaceae family that also includes the genus Cannabis. In beer hops provide bitterness to balance the sweetness of malt sugars, as well as flavors, aromas, resins that increase head retention, and antiseptics to retag spoilage. I have the following comments on the submitted manuscript:

1          1. The varieties are not characterized (f. e. aroma descriptors for hop cultivars).

  1. The amount of essential oils in cones is not listed (f. e. levels of alpha acids and essential oils vary among varieties giving them different characteristics that make them suitable for specific styles of beer and ale).
  2. Table 2 needs more interpretation and searching for differences in aromatic and taste substances in individual varieties (f. e. high alpha acid (bittering) types: (Alpha acids‐%‐ contribute to the bitter flavor of beer).
  3. It’s necessary to use more references for discussion, with an emphasis on world production of this specific crop – traditional & current usage.
  4. The conclusion is too broad. It’s necessary to state more important facts from the research conducted. Prospectively, it’s necessary to add information about the next research project, its aim, and its relation to product production technology, for example, beer.

 Minor editing of English language required Minor editing of English language required

Author Response

(The authors gave the same response as above.)

Reviewer 3 Report

Reviewer’s Comments

Title

The length of the title is long. Shorten the title within 15 words.

Abstract

What is the significance of this study? Why the content of those components were selected?

The abstract related to the description should be re-written since it has no statistical data to support your conclusion. Try to use quantitative figures to describe your results.

1. Introduction

In this section, there is no literature about the progress on the effect of cropping systems on hop quality. It is obvious that the literature review to introduce the background and the progress in the current research is not sufficient. Very few references were cited to conclude your results which cannot convince the readers that the research on this aspect is rare in Brazil. Therefore, please supplement at least 10 references at home and abroad to support your conclusions.

Besides, what is your novelty of the research compared with the previous studies. What is your hypothesis of your study? Please address those concerns in your revision.

2. Materials and Methods

Line 66: cite a reference since your statement was according to Koppen.

Line 69-71: present the long-term annual mean data of min and max air temperature, and precipitation.

Line 73-77: what do you mean of phytosanitary control? Also supplement the description of your replicates of your treatments. What was the area of your plots?

Line 79-88: supplement the name of producers, the cities and countries for each type of fertilizers.

Line 104-126: when were the hops samples collected? How did you pre-process those samples in the laboratory? More details about the plants leaves sampling should be presented here.

3. Results

Line 158-199: Is there any variety × cropping system interaction in your study? Please explain it in your revision.

Table 1

Why several data were missing in the Table?

4. Discussion

The discussion of the current study compared with the others studies seems difficult to understand due to a lack of enough treatments and more replications in experimental design. The statement in this section insufficiently support the conclusions from the authors.

5. Conclusions

Conclusions are fine.

Use abbreviations of names in Author Contributions.

References are in accordance with the guidelines.

Moderate polishing of English language was needed.

Author Response

(The authors gave the same response as above.)

Reviewer 4 Report

1.       The manuscript has ample scope for improvement in language and grammar. Typographical errors may be eliminated by a careful reading of the masnuscript by a native English speaker. Some of the points in this regard (only for example) include:

a.          Line 26: Delete one of the two “in”s

b.          Line 33: Change “use” to “uses”

c.          Line 35: Correct “biosynthesizes” to “biosynthesize”

d.          Line 36: Correct “others” to “other”

e.          Line 42: Correct “its” to “they”

f.           Line 45: Remove “The” before “xanthohumol”

g.          Line 50: Add “is” between “this” and “because”

h.          Line 51: Delete “as know,”

i.           Line 53: Correct “genetic” to “genetics”

j.           Line 79: Correct “Fertilizations were performed” to “Fertilization was done”

k.          Line 83: Correct “tecobuconazol” to “tebuconazole”

l.           Line 88: Correct “was” to “were”

m.       Line 91: Please refer to point ( j) above

n.          Line 99: Correct “per-formed” to “performed”

2.      It seems that soil testing was not done. If done, please provide the relevant data.

3.      Please indicate whether chemicals, microbials, organic concoctions and fertilizers used in the experiment were under recommended package of practices for the crop and the region. If yes, provide suitable references. If no, justify their appropriateness briefly in one or two sentences.

Please refer to the attached file

Author Response

(The authors gave the same response as above.)

Reviewer 5 Report

This research is on climate adaptation and the possibility of cultivating hops in tropical regions of Brazil. The research is referred to as Tropicalization, but only the chemical profiles and essential oils are reported. In climate adaptability  study of a crop or cultivation system tests, the first important parameters are  growth and biomass, followed by yield and then chemical composition. I recommended to reported growth and yield parameters.

Two cultivation methods, conventional and organic, have been compared in the treatments in main factor. This require a split plot design and analysis but is not reported in results.

The abbreviations used for cultivars and cultivation methods in the tables should also be mentioned in the Methods section of the article.

The most important compounds in hops, alpha acids, beta acids, and xanthohumol, are not significant. Looking at the high standard deviations in table 2, it seems that there high experimental error. Since all these compounds have been measured by the UHPLC-128 PAD/UV/Vis method, it appears that the measuring method was not standardized or the machine had issues. All the data measured by CG-MS and presented in Table 1 have very low standard deviations. If there were no differences in the values, the same high standard deviations should be seen in Table 1.

Author Response

(The authors gave the same response as above.)

Round 2

Reviewer 1 Report

After reading the manuscript I make the considerations.

Abstract: Clear and straightforward, but it would be nice to have an introductory sentence about importance culture. In practice, would there be any important difference to highlight for the market?

 Introduction. very clear and straightforward, I would just like to add something about conventional and organic types of agriculture and why to compare them as this is still not very clear.

 in item 2.3.1. The chemical composition of the essential oil is very long and difficult to read. Break the paragraph into several smaller paragraphs to make it easier to understand.

 in item 2.3.2. Quantification of alpha acids, beta acids and xanthohumol by an ultra-high efficiency liquid chromatograph coupled to a UV/Vis spectrophotometer (UHPLC-PAD/UV/Vis) it is too long and difficult to read, break the paragraph into several less to make it easier to understand.

 Line 150. using Minitab statistical software (reference);

 Figure 1. improve the legend.

Discussion. I believe the discussion itself was missing, not the explanation of each observed component. What are the differences between the types of management, why them, if the cultivation is not so common in the country, did the management used allow the production of fruits chemical composition of the essential oil in the expected standard? What could this mean for the country's market and industry?

 Line 255. In organic system (and what is the value in conventional systems, by way of comparison);

 Conclusion. the conclusion should respond to the objectives only. The cultivation of hop in Brazil is a reality, but there is still a lot of research to be done in order to contribute to the tropicalization of the species, which is essential for continuity and finally establishment of culture in the country. (I would replace this paragraph with something .

Moderate editing of English language.

Author Response

Dear reviewer,

Thank you for your time and for your valuable suggestions, all were considered and the necessary corrections were made.

Kind regards.

Reviewer 3 Report

Abstract

Introduce the research background and indicate your the significance of this study. The reason why the content of those components were selected is still not presented. Quantitative figures were used but it still needs more data to describe your results.

Introduction

Do not repeat the findings in literature review. Present comprehensive comparison between your current study and the previous findings. What is your novelty of the research compared with the previous studies. What is your hypothesis of your study?

Materials and Methods

No sense about the 2 x 5 subdivided plot. What is your unit for the area? What is the producer of Sisvar software?

Results

Authors did not reply the problem about the variety × cropping system interaction in your study.

Discussion

The authors did not fully address in their responses.

Abbreviations of names were still not used in Author Contributions.

Abbreviations of journal names in references are not correct. Further revision is needed.

Author Response

(The authors gave the same response as above.)

Reviewer 4 Report

Although the authors took professional service for improvement, some minor text editing regarding the language and grammar of the manuscript may be done.

Author Response

Dear reviewer, 

On behalf of all the co-authors I would like to thank you for your time and for your valuable suggestions.

It is an honor share these findings in such prestigious journal.

We are certain that our research can contribute to relevant information on this field of study.

Kind regards.

Reviewer 5 Report

Dear authors,

Two devices were used to measure the biochemical traits in hops. The traits measured by the first instrument, GC-MS, have low standard deviations and are statistically significant. However, all traits measured by the second method with the HPLC device have high standard deviations, and thus, this variance cannot be attributed to the field conditions or genotype. If this were the case, at least some traits measured by GC-MS should have a high standard deviation. I also accept the validity and accuracy of the data reported in this article. The main problem is with the measurement method or non-standardization of the device. Therefore, the article is not acceptable based on these data, and I cannot accept the authors' argument.

Author Response

Dear Reviewer,
Reviewing the original data we realized that there are some outlines that caused the high standard deviation.
If you give us the opportunity to work with the data, I'll be happy to make this work.
We await your reply.
Best regards.

Round 3

Reviewer 1 Report

After reading the manuscript entitled: Tropicalization of Humulus lupulus L. (Cannabaceae): Chemical compositions of varieties grown under organic and conventional cropping system in subtropical conditions, Brazil, by Gabriel Cássia Fortuna, Caio Scardini Neves, Olivia Pak Campos, Jordany Aparecida Oliveira Gomes, Júlio César Rodrigues Lopes Silva, Amauri Alves Souza, Cristiano Soleo de Funari, Márcia Ortiz Mayo Marques, Filipe Pereira Giardini Bonfim I make some considerations.

The authors were attentive to my observations and suggestions, which improved my understanding of the manuscript.

Author Response

(The authors gave the same response as above.)

Reviewer 3 Report

The manuscript has improved greatly. I suggest accepting the MS for publication.

Author Response

(The authors gave the same response as above.)

Reviewer 5 Report

Thank you for giving me the opportunity to review the revised manuscript. I apologize for the delay in my response.   I would like to note that the authors have not made any changes to the tables or data presented in the manuscript. However, they have made efforts to present the results with less emphasis on the section with higher standard deviation (SD). The authors have also provided additional information regarding the pre-analysis testing of the HPLC device and suggested that the reported high SD values may be due to environmental effects or inherent variability in the plant material, which is discussed in the relevant section of the manuscript.   I am pleased to inform you that the authors have made significant improvements and addressed most of the concerns that I raised in my previous review. After careful consideration, I  recommend that the manuscript be accepted for publication.   Thank you again for allowing me to review this manuscript, and please let me know if you have any questions or require further clarification.

Author Response

(The authors gave the same response as above.)
